# Evaluation of Everolimus Activity against *Mycobacterium tuberculosis* Using In Vitro Models of Infection

**DOI:** 10.3390/antibiotics12010171

**Published:** 2023-01-13

**Authors:** Delia Mercedes Bianco, Flavio De Maio, Giulia Santarelli, Ivana Palucci, Alessandro Salustri, Giada Bianchetti, Giuseppe Maulucci, Franco Citterio, Maurizio Sanguinetti, Enrica Tamburrini, Michela Sali, Giovanni Delogu

**Affiliations:** 1Dipartimento di Scienze Biotecnologiche di Base, Cliniche Intensivologiche e Perioperatorie—Sezione di Microbiologia, Università Cattolica del Sacro Cuore, 00168 Rome, Italy; 2Dipartimento di Scienze di Laboratorio e Infettivologiche, Fondazione Policlinico Universitario “A. Gemelli”, IRCCS, 00168 Rome, Italy; 3Department of Neuroscience, Biophysics Section, Università Cattolica del Sacro Cuore, 00168 Rome, Italy; 4Renal Transplant Unit, Fondazione Policlinico Universitario A. Gemelli, IRCCS, 00168 Rome, Italy; 5Dipartimento di Sicurezza e Bioetica, Sez. Malattie Infettive, Università Cattolica del Sacro Cuore, 00168 Rome, Italy; 6Mater Olbia Hospital, 07026 Olbia, Italy

**Keywords:** latent tuberculosis infection, host-directed therapies, Everolimus, mTOR inhibitors

## Abstract

Even though Everolimus has been investigated in a phase II randomized trial as a host-directed therapy (HDT) to treat tuberculosis (TB), an oncological patient treated with Everolimus for a neuroendocrine pancreatic neoplasia developed active TB twice and a non-tuberculous mycobacterial (NTM) infection in a year and a half time span. To investigate this interesting case, we isolated and genotypically characterized the *Mycobacterium tuberculosis* (*Mtb*) clinical strain from the patient and tested the effect of Everolimus on its viability in an axenic culture and in a peripheral blood mononuclear cell (PBMCs) infection model. To exclude strain-specific resistance, we tested the activity of Everolimus against *Mtb* strains of ancient and modern lineages. Furthermore, we investigated the Everolimus effect on ROS production and autophagy modulation during *Mtb* infection. Everolimus did not have a direct effect on mycobacteria viability and a negligible effect during *Mtb* infection in host cells, although it stimulated autophagy and ROS production. Despite being a biologically plausible HDT against TB, Everolimus does not exert a direct or indirect activity on *Mtb*. This case underlines the need for a careful approach to drug repurposing and implementation and the importance of pre-clinical experimental studies.

## 1. Introduction

Tuberculosis (TB) remains a major global health concern, being one of the leading causes of death from a single infectious agent [1]. Fully susceptible Mycobacterium tuberculosis (*Mtb*) is usually treated with a multidrug pharmacological regimen, consisting of daily dosing of four drugs, for a period ranging from 4 to 8 months [2].

The emergence of multidrug-resistant tuberculosis (MDR-TB) and extensively drug-resistant tuberculosis (XDR-TB) and the high rate of patient non-compliance to therapy [3] due to the duration made the identification of novel potential treatment strategies a priority. However, drug development is a long and cumbersome process that may require decades before a new drug can be introduced into current regimens [4,5]. Drug repurposing is seen as an opportunity to shorten this timeframe, though properly run randomized clinical trials are needed to assess the efficacy of these new drugs against TB [6].

Ref. [7] Thanks to a better understanding of TB pathogenesis, a new approach is to focus on factors that can influence pathogen survival; drugs acting on the host rather on the bacterium itself are identified as host-directed therapies (HDTs) [8,9]. Many studies have been and are being carried out to identify approved drugs with promising therapeutical effects for TB [8,10], though very few solid clinical trials have been completed.

Among the proposed molecules “to repurpose” against TB is Everolimus, a molecule that exerts immunosuppressant activity by inhibiting the mTOR pathway and blocking growth-factor driven T-cell proliferation [11] that has been described as exerting both anti-mycobacterial and immunomodulatory activities [12,13]. Everolimus is currently used at low doses (3–5 ng/mL) in renal transplantation as an immunosuppressive drug in combination with low dose Tacrolimus. The largest prospective, randomized, clinical study ever designed in transplantation, enrolling 2037 kidney transplant recipients, has shown non inferior immunosuppressive efficacy, with respect to the combination of tacrolimus and mycophenolate, but significantly superior activity in reducing viral infections, namely reducing the incidence of CMV and BK virus infections [14]. When used in transplant clinical practice, Everolimus therapeutic concentrations range from 3 to 8 ng/mL. Within this clinical range, no increased or decreased incidence of TB has been reported in transplant recipients.

Experimental findings highlighted that both autophagic and non-autophagic mechanisms contribute to *Mtb* killing after treatment with Everolimus alone or in combination with first line drugs [12,15]. Indeed, Everolimus was demonstrated to impair *Mtb* growth in axenic culture, to reduce *Mtb* viability in an in vitro infection model, and to increase ROS production [12]. A phase II randomized clinical trial, the TBHDT (NCT number: NCT02968927), showed that Everolimus does not significantly improve clinical recovery when administered as an adjunct therapy during the standard anti-TB regimen [16].

Interestingly, a patient in our institution treated with Everolimus for a neuroendocrine pancreatic neoplasia developed active TB twice and a non-tuberculous mycobacterial (NTM) infection in a time span of a year and a half, despite appropriate antitubercular treatment. This case was in line with some anecdotal cases of TB development or reactivation during Everolimus therapy [17,18], but in apparent contrast to the potential usefulness of Everolimus as a HDT for TB. Thus, we decided to further investigate the molecule in vitro using four different concentrations.

## 2. Results

### 2.1. Case Report Description

The patient is a 62-year-old East Asian male with a history of pancreatic neuroendocrine tumour diagnosed in 1995 and treated with pancreatectomy and splenectomy. Since the patient was referred to our centre only in August 2018, the clinical information from 1995 to 2018 are partial. Briefly, in 2003, the patient developed hepatic metastasis from the same primary tumour and was treated with a chemotherapeutic regimen of cisplatin, etoposide, and FOLFIRI from 2004 to 2016, with metastasectomy of the IV liver segment and lesion chemoembolization in 2009. In 2016, he started therapy with Everolimus. At the end of 2017, an *Mtb* infection was first diagnosed through a positive tuberculin skin test (TST) and interferon gamma release assay (IGRA—QuantiFERON TB Gold Plus) for which he was treated with a combination of rifampicin and isoniazid until April 2018.

In June 2018, the patient underwent a thoracic X-ray due to recurrent fever, fatigue, lack of appetite, and dry cough for more than one month. Imaging revealed lung consolidation and treatment with amoxicillin/clavulanic acid and ciprofloxacin was started based on a radiological diagnosis of pneumonia. Therapy with Everolimus was suspended.

In August 2018, the patient underwent a second radiological exam revealing a 5 cm consolidation in the perihilar area of the right lung with evidence of excavation, for which he was referred to the emergency department. He was placed in airborne infection isolation and an infectious diseases consultant recommended collecting sputum samples to search for mycobacteria and to start therapy with piperacillin/tazobactam.

The patient was then referred to our centre where a diagnosis of active TB was made based on X-ray imaging, sputum analysis, and clinical findings. Sputum analysis revealed a fully sensitive *Mtb* strain (*Mtb* GM2659) that was treated with standard quadruple therapy (2HRZE/4HR).

The patient was assessed monthly to closely monitor treatment response, with sputum smear and cultures, and for effective counselling. A PET-CT performed in November 2018 showed neoplastic disease progression, for which the patient was again started on Everolimus.

In May 2019, the patient complained of blood-stained sputum and a second diagnosis of active TB was made, as well of *Mycobacterium fortuitum* infection, based on sputum analysis and molecular assays. He was continued on a treatment with rifampicin and isoniazid until July 2019 and tested negative on sputum smear and culture in October 2019. Strict follow-ups at one and six months were scheduled. The complete clinical case is schematically described in Figure 1.

### 2.2. Experimental Results

#### 2.2.1. Characterization of the Clinical Isolate

*Mtb* clinical isolate GM2659 was phenotypically and molecularly characterized following microbiological diagnosis (August 2018). GenoType^®^ MTBDR assay, phenotypic MGIT, and proportional DST assays showed no resistance determinants. Furthermore, the genome of GM2659 was sequenced and the strain classified as belonging to the EAI “Manila” clade (lineage 1). Whole genome sequencing confirmed no other unknown mutations associated with genes inducing drug resistance. *Mtb* GM2659 and *Mtb* H37Rv showed a comparable growth rate in axenic culture (Appendix A).

#### 2.2.2. Evaluation of the Direct and Indirect Activity of Everolimus on Mtb

As Everolimus has been suggested as a potential HDT against *Mtb* infection, having both direct anti-mycobacterial activity and an ability to modulate host immune responses [12], we first evaluated the direct anti-mycobacterial activity of Everolimus in liquid axenic culture against the *Mtb* GM2659 and *Mtb* H37Rv strains at the recommended therapeutic range for kidney transplantation (minimal: 3 ng/mL = 3.13 nM; maximal 8 ng/mL = 8.35 nM). As reported in Figure 2A,B, Everolimus did not affect the mycobacterial growth rates (*p > 0.05*) of either strain. Additionally, we tested two other mTOR inhibitors, Temsirolimus and Sirolimus, at the minimal and maximal recommended therapeutic concentrations, observing no differences in terms of CFUs compared to untreated control.

To measure the indirect activity of Everolimus against *Mtb*, that is the activity as a HDT, we evaluated the impact of Everolimus on *Mtb* replication and survival in a PBMC infection model [9]. PBMCs were infected with the *Mtb* H37Rv reference strain and the clinical isolate *Mtb* GM2659, then treated starting at four days post infection with Everolimus (3.13 nM and 8.35 nM), with CFUs assessed 8 days post infection. Everolimus administration resulted in a non-significant reduction in CFU counts for *Mtb* H37Rv and for the *Mtb* GM2659 clinical strain (Figure 2C). To corroborate these findings, different concentrations (ranging from 10^6^ CFUs/mL to 10^4^ CFUs/mL) of *Mtb* H37Rv were plated on 7H11 solid medium containing Everolimus at increasing concentrations (1 ng/mL, 3 ng/mL, 8 ng/mL, and 20 ng/m) or INH. On the other hand, the same strain was used to challenge PBMCs that were treated with Everolimus (1 ng/mL, 3 ng/mL, 8 ng/mL, and 20 ng/m) 4 days post infection (Appendix A). Mycobacterial growth on solid medium and CFU counting did not show any differences when compared to untreated controls. Taken together, these results suggest that Everolimus administration at commonly used therapeutic concentrations does not have a direct effect on mycobacteria viability nor an effect during *Mtb* infection in host cells.

#### 2.2.3. Autophagy and ROS Production

The impact of Everolimus on the autophagic flow has been described as a key factor in controlling *Mtb* infection [12]. To investigate the ability of Everolimus to modulate autophagy in *Mtb*-infected macrophages, THP-1 cells transfected with mRFP-GFP-LC3B were infected with the *Mtb* H37Rv and GM2659 strains, and the autophagic flux was evaluated using confocal microscopy pH-imaging of the autophagic intermediates measurement (AIPD) [19]. As control, we included two previously characterized *Mtb* strains: one belonging to the modern Euro-American superlineage (*Mtb*^H3^) and the second to the ancient East African India superlineage (*Mtb*^EAI^), which served as a control for the *Mtb* GM2659 strain isolated from the patient [20]. While activation of the autophagic process does not systematically correlate with concomitant restriction of *Mtb* intracellular replication, modern and ancient strains are differentially affected by this process [9,20]. Using mRFP-GFP-LC3B THP-1 macrophages, any change in the autophagy pathway determines a variation in AIPD allowing a qualitative and quantitative estimation of the autophagic flux: an increase in the *F_G_*/*F_R_* ratio corresponds to an increase in autophagosomes (green/yellow signal), whereas lower *F_G_*/*F_R_* values indicate autolysosome generation and a decrease in the autophagic intermediate pH (red signal due to GFP quenching) (Figure 3A). As illustrated in Figure 3B,C, the autophagic flux was comparable for all the untreated mycobacterial strains except for *Mtb*^EAI^, in which autophagosomes appeared slightly increased. Following Everolimus administration, an increase in the number of autophagosomes was detected in all samples except for *Mtb*^EAI^ (Figure 3B,C). Intriguingly, no significant variations in pH were found (Figure 3D). Although we cannot exclude a rapid elimination of the intracellular *Mtb*, no increase in autolysosomes was detected suggesting blockage of the autophagic process at an early step of infection (Figure 3B–D).

Previous studies linked the observed anti-mycobacterial effect of Everolimus to its ability to induce the formation of reactive oxygen species (ROS) [12]. Hence, we measured ROS in THP-1 macrophages previously infected with *Mtb* H37Rv, GM2659, *Mtb*^H3^, and *Mtb*^EAI^ strains following Everolimus administration at maximum therapeutic concentration. After infection, cells were marked with CellROX reagent (ThermoFisher) which measures oxidative stress in live cells. This compound is a DNA dye weakly fluorescent in the reduced state, which exhibits bright fluorescence upon oxidation (Figure 4A). However, in our experimental settings, we did not observe any significant increase in ROS production (Figure 4B–C). Taken together, these findings suggest that *Mtb* persists in early autophagosomes with no or only a minor shift versus autolysosomes, regardless of Everolimus treatment.

#### 2.2.4. Long Term Effects of Everolimus

The previously discussed experiments showed that Everolimus has no effect on mycobacterial replication or autophagy modulation during the early stage of *Mtb* infection, but we could not exclude potential effects of Everolimus at late phases of the infection. For this reason, we infected PBMCs with the *Mtb* strains (*Mtb* H37Rv, GM2659, *Mtb*^H3^, and *Mtb*^EAI^) and treated them with Everolimus 4 days post infection, as previously indicated. Mycobacterial viability was assessed 12 days post infection by CFU measurement. No decrease in CFUs was observed when PBMCs were infected with the *Mtb* H37Rv reference strain, as well as with the *Mtb* clinical strains: *Mtb*^H3^, GM2659, and *Mtb*^EAI^. (Figure 5). These results confirm that Everolimus does not help to contain *Mtb* isolates in infected PBMCs.

## 3. Discussion

An experimental paper by Ashley et al. [12] reported that Everolimus has been proposed to exert a strong anti-mycobacterial activity in axenic culture and during *Mtb* infection of PBMCs by regulating autophagy and inducing ROS production [12].

Despite these promising results on the use of mTOR inhibitors as a HDT against TB, the administration of these molecules has been widely described as a risk factor for TB reactivation [17,18,21,22]. Indeed, both Fijalkowska-Morawska et al. [17] and Jeon et al. [18] described cases of active TB a few months into Everolimus treatment, raising concerns about the application of this pharmaceutical intervention. Moreover, in a phase II randomised controlled trial, adjunctive use of Everolimus only resulted in a non-statistically significant improvement of FEV1 (forced expiratory volume at 1 s) at six months when compared to standard of care [16]. No effect was observed in sputum culture in solid and liquid media at day 56 and 180, or in function at day 56 and 540 [16]. 

The clinical case we described prompted us to investigate the potential activity of Everolimus against *Mtb* using several experimental models and four different concentrations.

In our settings, Everolimus, when administered at standard therapeutic concentrations, which by the way are higher than those previously used by Ashley et al. [12], does not show any direct effect on *Mtb*, and neither do its analogues Temsirolimus and Sirolimus. We decided to use four different concentrations of Everolimus (1 ng/mL, 3 ng/mL, 8 ng/mL, and 20 ng/mL) based on previous papers testing Everolimus as a HDT [12], the therapeutic range for kidney transplantation, and the probable plasma concentration in our patient [23].

We also demonstrate that Everolimus does not exert a significant direct or indirect activity against *Mtb* using a panel of clinical strains, including the reference H37Rv, the strain GM2659 isolated from our patient, and the well characterized *Mtb* strains belonging to modern and ancient lineages [20]. Our findings do not lend support to the previous observations indicating that Everolimus shows anti-TB activity and highlight the pitfalls of previous studies using only laboratory strains when testing new compounds [12].

Autophagy is a process in which cells degrade dysfunctional and unnecessary cellular components, including cellular constituents and intracellular bacteria sequestered into autophagosomes that are then transported to endosomes or lysosomes to become autophagolysosomes, in which hydrolase enzymes degrade their contents [19]. The role of autophagy during *Mtb* infection remains controversial and still elusive. Previous reports observed that enhanced autophagy was related to enhanced *Mtb* killing [15]. Conversely, recent data demonstrate that not only inhibition of autophagy may impair *Mtb* intracellular replication [9], but *Mtb* strains of modern superlineages may also induce autophagy to promote *Mtb* intracellular replication [20].

In our experiments, Everolimus was able to induce autophagy flux, but we did not observe a reduced *Mtb* intracellular replication in either modern or ancient clinical strains. Intriguingly, our results suggest that mycobacteria can inhibit the late phases of autophagy progression, with early autophagosomes serving as a protected niche for replication [9,20]. This is in line with recent findings showing that ancient strains can also modulate autophagy [20]. Thus, even though the autophagy pathway is overactive in Everolimus-treated cells, we could not measure a significant impact on *Mtb* viability.

As our patient was affected by a neuroendocrine neoplasia for more than 20 years, a disease known to negatively affect a patient’s metabolic and nutritional status, and was on different chemotherapeutic regimens, it is reasonable to assume that the patient analysed in this study had a higher vulnerability to adverse health outcomes on different chemotherapeutic regimens (Figure 1) [24]. We can hypothesize, as previously postulated by Vale [25], that HDTs may have a different effect on an already compromised host, suggesting paying particular attention before administrating these not-conventional treatments

Moreover, individual susceptibility to infection is thought to be linked to genetic variation in key genes encoding for proteins involved in the adaptive and innate immune response [26,27]. The frequency of genes encoding for these proteins varies among different populations, to the point that some proteins are said to have a population-specific structure [28,29].

Many clues suggest that there have been some reciprocal adaptations of *Mtb* and the human host; some lineages have been defined as ‘specialists’ for their persistence in specific host populations [30]. This kind of genetic variant could also affect patients’ response to HDTs, given the variety and complexity of their mechanisms of action. In our patient, we were unable to exclude the presence of host gene variants that may be associated with increased susceptibility to mycobacterial infection or to the decreased effectiveness of Everolimus against *Mtb*. This may be relevant given that the patient suffered not only *Mtb* infection, but also subsequent infection by *Mycobacterium fortuitum*.

In conclusion, the experimental evidence obtained in our study does not show a direct activity of Everolimus against *Mtb* and does not lend support to the indirect activity of Everolimus as a HDT against *Mtb*. The biological effects of Everolimus on the autophagy process and on ROS production did not result in a significant anti-*Mtb* activity, regardless of the genetic features of the *Mtb* strains. Our findings suggest caution before considering Everolimus a valid HDTs against *Mtb* and are a prompt for more studies.

## 4. Materials and Methods

### 4.1. Sample Collection and Processing

Clinical samples, collected according to the institution’s guidelines, were investigated for the presence of *Mtb* infection. Sputum specimens were processed by the conventional *N-acetyl*-L-*cysteine-sodium hydroxide* decontamination method, and the sediment was re-suspended in PBS. The suspension was used to prepare: (a) Lowenstein–Jensen cultures (LJ); (b) Mycobacteria Growth Indicator Tube (MGIT) liquid (Becton Dickinson Diagnostic Systems) cultures; (c) smears for microscopic examination by *Ziehl–Neelsen* staining; and (d) molecular assays for mycobacteria detection and identification [31].

### 4.2. Routine Standard Culture–Based Drug Susceptibility Tests

A drug susceptibility test (DST) was performed using the Bactec MGIT 960 system (Becton Dickinson Diagnostic Systems), with the following drugs: streptomycin (STR), isoniazid (INH), rifampin (RIF), and ethambutol (EMB) from the M960 SIRE kit, following the manufacturer’s procedures. The M960 system conveyed the final interpretation and susceptibility results automatically.

Phenotypic DSTs were performed using Middlebrook 7H10 (Difco) agar medium containing standard critical drug concentrations of INH, RIF, EMB, STR, Ethionamide, Capreomycin, Kanamycin, Ofloxacin and Para-amino salicylic acid. 7H10 drug-containing plates were inoculated with a suspension of mycobacteria. Growth was detected 3–4 weeks later and was compared to the growth on control plates. The ratio between the number of colonies in the medium containing the anti-TB drugs and that of the control plates was calculated.

### 4.3. Molecular Assay to Detect Mycobacteria in Clinical Specimens and to Identify Resistance Determinants

The Anyplex *MTB*/NTM real-time detection assay (Seegene) was performed [32]. An aliquot of the decontaminated samples was firstly boiled, centrifuged, and finally used as the PCR template for the assay as suggested by manufacturer’s instructions. Amplification was performed on a CFX96TM real-time PCR system (Bio-Rad). Result interpretation was performed according to threshold and cut-off values outlined by the manufacturer. GenoType^®^ MTBDR assay (Hain LifeScience GmbH) or a nucleic acid amplification test (NAAT) Xpert MTB/RIF (*Cepheid*) was used to detect INH and RIF resistance or to detect *Mtb* and non-tuberculous mycobacteria (NTM) and RIF resistance, respectively.

### 4.4. Bacterial Strains and Growth Conditions

GM2659 clinical isolate, *Mtb*^H3^, *Mtb*^EAI^ [20], and *Mtb* H37Rv were grown at 37 °C in Middelbrook 7H9 broth medium (Difco), supplemented with 10% ADC (Becton- Dickinson), 0.2% glycerol (Sigma), and 0.05% Tween 80 (Sigma), and stocked at −80 °C after adding 20% glycerol [32]. The mycobacterial growth rate was assessed using MGIT liquid culture (Becton Dickinson) as described above [31].

### 4.5. Whole Genome Sequencing and Bioinformatic Analysis

DNA of the GM2659 clinical isolate was extracted from liquid culture using the CTAB method [33] and the library was prepared using Nextera DNA Flex Library Prep Kit (Illumina) and Nextera™ DNA CD Indexes (Illumina) according to Illumina’s instructions. FastQ sequences were uploaded onto PhyResSE web software to detect resistance determinants [34].

### 4.6. Culture–Based mTOR Inhibitors Drug Susceptibility Tests

*Mtb* H37Rv reference strain and GM2659 clinical isolate (≈1 × 10^5^ CFUs/mL) were incubated with the minimal and maximal therapeutic concentration of Everolimus (3 and 8 ng/mL, respectively), sirolimus (5 and 15 ng/mL, respectively) and temsirolimus (585 and 2400 ng/mL, respectively) (Sigma) [35,36]. Untreated cultures and samples incubated with INH (0.2 ng/mL) were used as controls. Each experiment was carried out in 7H9 medium enriched with 10% ADC and 0.05% Tween80, as described above. Bacteria were seeded in sterile 5 mL tubes and incubated at 37 °C for 15 days. An aliquot of each suspension was serially diluted and plated on 7H11 containing 10% OADC which was incubated in standard atmospheric conditions.

### 4.7. Cell–Based Drug Activity Test

Peripheral blood mononuclear cells (PBMCs) obtained from healthy donors were seeded in 48-well plates at final concentration of 1.2 × 10^6^ cells/mL. PBMCs were infected with *Mtb* H37Rv, GM2659 *Mtb*^H3^, and *Mtb*^EAI^ at MOI 1:10 and incubated at standard atmospheric conditions. Four days post infection, Everolimus at minimal (3 ng/mL = 3.13 nM) and maximal (8 ng/mL = 8.35 nM) therapeutic concentrations was added to infected cells. At day 8 post infection, and again at day 12, for all the strains, supernatant was removed to eliminate extracellular bacteria, and colony-forming units (CFUs) were determined by harvesting infected cells with 0.1 mL of sterile Triton X-100 (Sigma-Aldrich). Serial dilutions were performed before plating on 7H11 solid medium containing 10% OADC. Plates were incubated at 37 °C for 15–20 days.

### 4.8. Cell Cultures for Autophagy and ROS Evaluation

2-well chamber slides (Ibidi) were coated with Poly-D-Lysine for 1h and then used to grow wild-type human THP-1 cells and THP-1 EGFP-RFP-LC3. Cells were incubated in RPMI 1640 supplemented with glutamine (2mM) and 10% FBS in a humidified atmosphere (5% CO_2_ at 37 °C). Secondly, cells were treated with 20 nM of PMA (Sigma-Aldrich) for 24 h to induce their differentiation into macrophages and washed three times with PBS. Then, cells were infected with *Mtb* strains (*Mtb* H37Rv, GM2659 *Mtb*^H3^, and *Mtb*^EAI^) at a MOI of 1:1. One hour post infection, cells were treated with Everolimus at its maximum therapeutic concentration (8ng/mL). Finally, at five hours post infection, cells were washed with sterile warm PBS and fixed using 4% paraformaldehyde (PFA) for 30 min.

To assess ROS production, differentiated THP-1 cells, infected as previously described, were treated with CellROX green reagent (Thermofisher) (5 µM) four and a half hours post infection for 30 min. After incubation, cells were washed three times with sterile warm PBS and fixed using 4% PFA for 30 min.

### 4.9. Confocal Microscopy

Images were obtained using an inverted confocal microscope (Nikon A1 MP) equipped with an on-stage incubator (T = 37 °C, 5% CO_2_, OKOLAB). Internal photon multiplier tubes collected 16-bit images at 0.25 ms dwell time. mRFP-GFP-LC3 was excited by an argon-ion laser line (excitation wavelength, 488 nm; emission ranges 525/50 nm (*F_G_*), 595/50 nm (*F_R_*)). Detector gain values were kept fixed during the experiment. The pinhole was set to 1.2 A.U. Analysis of the images acquired was performed with ImageJ 1.41 (NIH). Autophagic intermediate pH distribution (AIPD) determination was obtained following Maulucci et al. [19]. Briefly, the *F_G_*/*F_R_* index was obtained by calculating the ratio between fluorescence emissions in the two detection ranges, upon sample excitation at 488 nm. By mapping *F_G_*/*F_R_* over the entire microscope scanning field, *F_G_*/*F_R_* images can be created with Image J; maxima of red and green channels, representing autophagy intermediates (“puncta”), were retrieved using the FIND MAXIMA plugin (ImageJ) and fluorescence intensity values were measured directly (ImageJ). Puncta without detectable EGFP fluorescence were minimized to <5% of the total number by setting adequate values for photomultipliers. At least 50 cells per sample were analysed to build the histogram. CellRoX fluorescence was quantified by excitation at 488 nm in the 595/50 nm emission range and measured using ImageJ. Fluorescence intensities and intensity ratio data were presented as mean ± SD, and differences were assessed using T-test. Values of *p* < 0.05 were considered statistically significant.

### 4.10. Data Analysis

Data were collected and organized using Microsoft Excel. Data were analysed by using GraphPad Prism software version 8 (GraphPad software). All experiments were performed in scientific duplicates and technical triplicates. CFU data were reported as mean plus SD and analysed by one-way ANOVA comparison tests followed by appropriate correction.

## Figures and Tables

**Figure 1 antibiotics-12-00171-f001:**
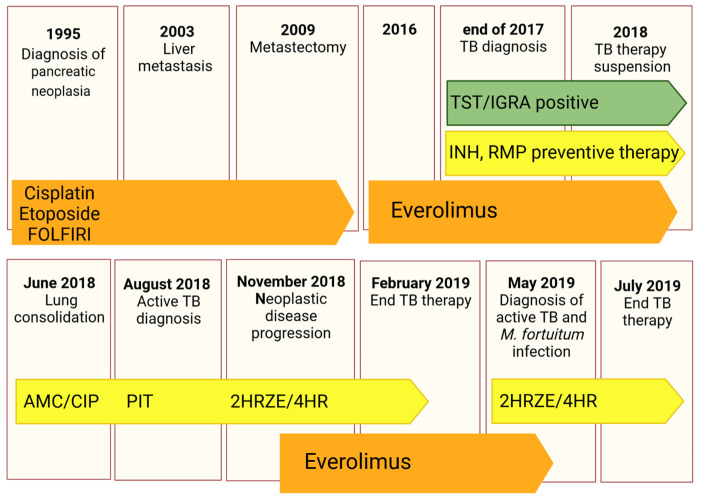
Schematic timeline reporting information on the clinical history of the patient, including microbiological examinations and treatments. Oncological therapies are reported in orange, while antibacterial ones in yellow and green boxes illustrate the immunological tests to diagnose *Mtb* contact. Antineoplastic drugs (Cisplatin, Etoposide, FOLFIRI regimen, and Everolimus), antibiotics (AMC: Amoxicillin/clavulanate; CIP: ciprofloxacin; and PIT: Piperacillin/tazobactam) and anti-TB treatments (INH: Isoniazid, RMP: Rifampicin, HRZE: Isoniazid/rifampicin/pyrazinamide/ethambutol, HR: isoniazid/rifampicin) are reported.

**Figure 2 antibiotics-12-00171-f002:**
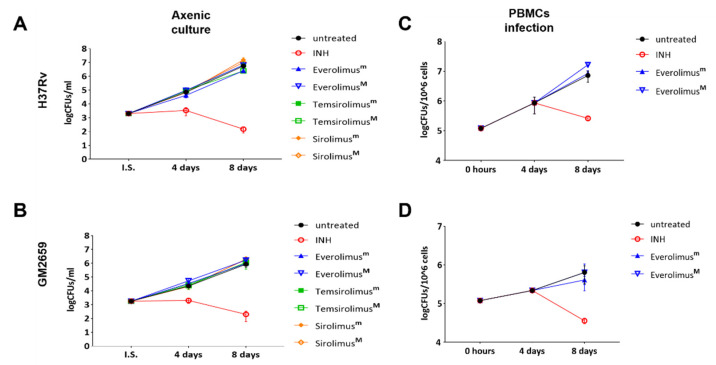
Anti-mycobacterial activity of Everolimus and analogue compounds Temsirolimus and Sirolimus on *Mtb* reference strain (**A**) and clinical isolate GM2659 (**B**). A suspension of mycobacteria, in 7H9 liquid medium supplemented with 10%ADC and 0.05% Tw80, was incubated with both minimal (m) and maximum therapeutic concentrations (M) of the selected compounds (Everolimus [3 and 8 ng/mL], Temsirolimus [585 and 2400 ng/mL], and Sirolimus [5 and 15 ng/mL]). An untreated suspension and a suspension containing Isoniazid (INH) were used as controls. Colonies forming units (CFUs) were evaluated at 4 and 8 days and reported as average ± SD in log10 scale. Everolimus effects on modulating the host response were tested in an ex vivo infection model. Peripheral blood mononuclear cells (PBMCs), isolated from heathy volunteers, were infected with a MOI of 1:10 with *Mtb* H37Rv (**C**) and GM2659 (**D**). Everolimus, at therapeutic concentrations (Everolimus^m^—3 ng/mL, Everolimus^M^—8 ng/mL), was added 4 days post infection. Colony-forming units (CFUs) were assessed at days 8 post infection and reported as average ± SD in log10 scale.

**Figure 3 antibiotics-12-00171-f003:**
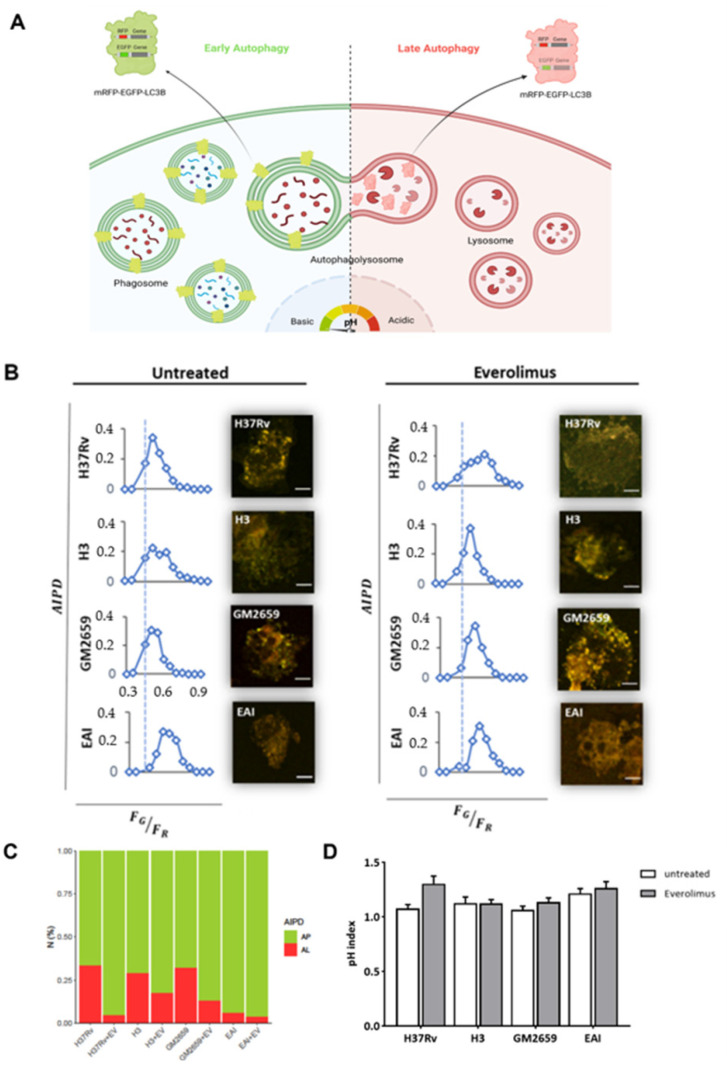
Schematic representation of the mRFP-EGFP-LC3B reporter system used to assess the impact of Everolimus on the autophagic process [19]. (**A**) THP1 cells constitutively express a recombinant LC3B linking mRFP and EGFP. When autophagosomes fuse with a lysosome the mRFP-EGFP-LC3B is lipidated, and it detaches from the autolysosome membrane. The signals from EGFP that were present within the inner autophagosome membrane are quenched in the autolysosomal environment. The number and pH of autophagic intermediates are reported as AIPD and as pH distribution of the number of autophagic intermediates per cell. AIPD shape and amplitude are sensitive to alterations in the autophagy pathway and allow a quantitative estimation of autophagic flux by retrieving the concentrations of autophagic intermediates. An increase in *F_G_*/*F_R_* ratio indicates an increase in autophagomes. A shift of AIPD toward low *F_G_*/*F_R_* values indicates that the pH of autophagic intermediates is shifting to acidic values and autolysosome formation. THP-1 RFP-LC3-GFP cells were infected with *Mtb* H37Rv, GM2659, *Mtb*^H3^, and *Mtb*^EAI^ at MOI 1:1 (**B**). Cells were treated with Everolimus^M^ 8ng/mL one hour post infection. Quantitative assessment of autophagic flux and pH variation were assessed five hours post infection. Reported in the graphs are the AIPD and pH distribution expressed as ratios *F_G_*/*F_R_* between the AIPD area of a fixed threshold value. Ratio between auophagosomes (AP) and auophagolysosomes (AL) is reported as a bar plot (**C**). pH variations (pH index) are represented as histograms (**D**). Data were analysed using two-way ANOVA with Bonferroni’s multiple-comparisons test.

**Figure 4 antibiotics-12-00171-f004:**
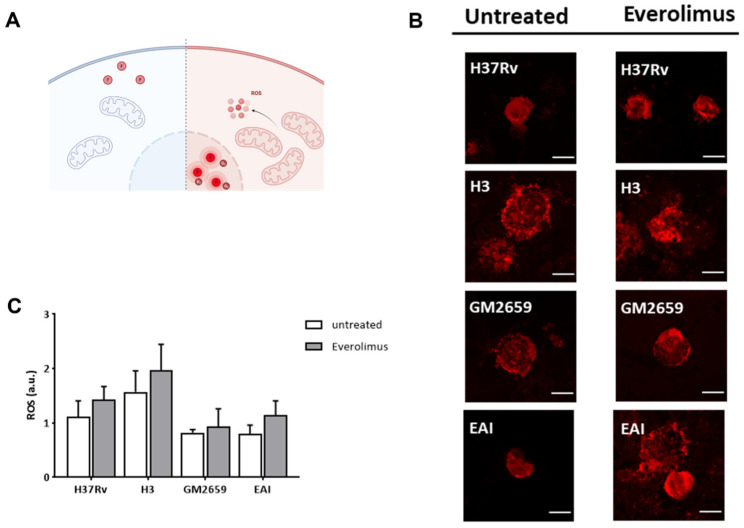
Schematic representation of the functioning of CellROX green reagent used to assess the impact of Everolimus on ROS production (**A**). CellROX reagent is non-fluorescent in a reduced state, while upon oxidation it binds to DNA principally in the nucleus and mitochondria, emitting a strong fluorogenic signal. Cells were treated with Everolimus^M^ 8 ng/mL one hour post infection. Oxidative stress was assessed five hours post infection. Cells were fixed with PFA and then imaged with a confocal microscope (**B**). ROS production was quantified and represented as barplot (**C**). Data were analysed using two-way ANOVA with Bonferroni’s multiple-comparisons test. Everolimus-treated cells had higher levels of ROS, although this difference was not statistically significant.

**Figure 5 antibiotics-12-00171-f005:**
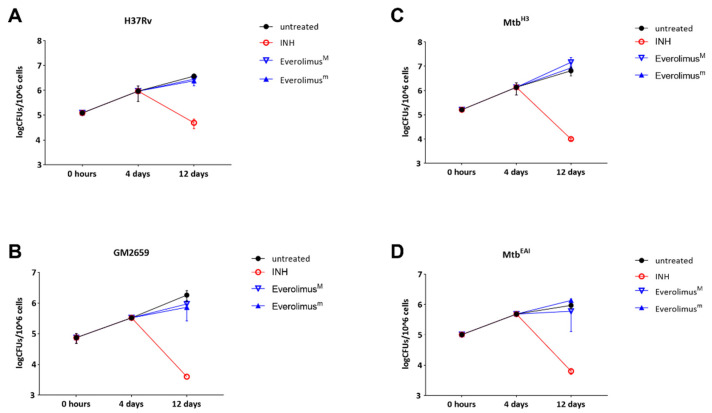
Evaluation of the long-term effect of Everolimus maximal and minimum therapeutic concentrations (Everolimus^M^ and Everolimus^m^) in the ex vivo *Mtb* infection model. Modern (*Mtb* H37Rv and *Mtb*^H3^) and ancient (GM2659 and *Mtb*^EAI^) strains were used to infect peripheral blood mononuclear cells (PBMCs) at MOI = 1:10. Everolimus^m^ (3 ng/mL), Everolimus^M^ (8 ng/mL), and INH (0.2 µg/mL) were added four days post infection. Colony-forming unit (CFU) analysis was performed 4 days post infection and 12 days post infection, and reported for *Mtb* H37Rv (**A**), GM2659 (**B**), *Mtb*^H3^ (**C**), and *Mtb*^EAI^ (**D**). Average ± SD of CFUs were represented in log10 scale.

## Data Availability

Data are available upon request to the corresponding author.

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
