# Peer review of "Evaluation of Everolimus Activity against Mycobacterium tuberculosis Using In Vitro Models of Infection"

_antibiotics, 2023, doi:10.3390/antibiotics12010171_

Round 1

Reviewer 1 Report (Previous Reviewer 2)

The authors response is satisfactory.

Although there are minor points need to address

1.    Cite the references to line 295-296 in the manuscript.

2.    Including the study aim in introduction section makes it understandable about the nature of the study. Although explaining it in the discussion section with greater details can make it very clear and avoid the misinterpretation.

Author Response

Reviewer 1

Although there are minor points need to address

  1. Cite the references to line 295-296 in the manuscript.

We thank the reviewer for his/her suggestion, we have included the missed references in the revised manuscript.

  1. Including the study aim in introduction section makes it understandable about the nature of the study. Although explaining it in the discussion section with greater details can make it very clear and avoid the misinterpretation.

We thank the reviewer for his/her suggestion. We have revised the sections and included the study aim in the first paragraph of the discussion section.

Reviewer 2 Report (New Reviewer)

Review 

The work presents an interesting, multistep study of the effect of Everolimus on Mycobacterium tuberculosis clinical strains using valuable in vitro analysis. As the search for new solutions for tuberculosis treatment is essential for the development of new drugs the topic of the research is valid. The obtained results do not give the complete answer and further research is required, but as standing in opposition to so far published papers and this paper would be interesting not only as the basis for further analysis of the Authors but also for the other research groups.

Generally, the study is well designed and interpreted, can arouse public interest, and can consider for publication, nevertheless, minor revision is required. 

The clinical case description is described widely, and the description of the treatment is broad. Nevertheless, this case is an isolated one, and it can not be treated as an indication of a general trend. Furthermore, it is described at the beginning as the important part of the manuscript, when the results are less trustworthy and statistically valid than the following in vitro test. Therefore appropriate comments or discussions should be added..

Author Response

The work presents an interesting, multistep study of the effect of Everolimus on Mycobacterium tuberculosis clinical strains using valuable in vitro analysis. As the search for new solutions for tuberculosis treatment is essential for the development of new drugs the topic of the research is valid. The obtained results do not give the complete answer and further research is required, but as standing in opposition to so far published papers and this paper would be interesting not only as the basis for further analysis of the Authors but also for the other research groups.

Generally, the study is well designed and interpreted, can arouse public interest, and can consider for publication, nevertheless, minor revision is required.

We thank the reviewer for the overall positive comment.

In the point-by-point rebuttal, we address the specific points raised by the reviewer.

The clinical case description is described widely, and the description of the treatment is broad. Nevertheless, this case is an isolated one, and it can not be treated as an indication of a general trend. Furthermore, it is described at the beginning as the important part of the manuscript, when the results are less trustworthy and statistically valid than the following in vitro test. Therefore appropriate comments or discussions should be added..

We thank the reviewer for his/her suggestion.

We have revised the introduction and discussion section to specifically address the comment raised by the reviewer.

Reviewer 3 Report (New Reviewer)

The manuscript submitted by Delia Mercedes Bianco et al reports a case study of a pancreatic neuroendocrine tumour patient diagnosed with the tuberculosis after taking Everolimus medication. The authors further performed in vitro analysis to look at the root cause reasons. This manuscript is well structured overall, but it needs minor modifications, that I have outlined in the comments below.

1.   Introduction section is very concise, please elaborate. 

2. Line 41-42: The sentence is incomplete and does not make sense to the readers. Please correct them.

3.  Line 45-48: Please add reference to this sentence.

4. Line 83: The authors have to explain in detail why the cancer drugs cisplatin, etoposide and FOLFIRI was stopped in 2016 and started a new drug Everolimus?

5.  Line 90: If Everolimus drug was suspended, what other anticancer drug was given to the patient for pancreatic neuroendocrine tumour?

6. What  dosage of Everolimus drug  was given to the patient? Did the authors used the same concentration for the in vitro studies?

7.   Please add references to the section 2.2.2.

8. Line 173-177: To look at the broader prospect, why the authors haven’t included another Asian Mtb strains as a control?

9.    Please add conclusion section.

10.  The authors have to proofread the manuscript, there are many grammatical errors.

Author Response

The manuscript submitted by Delia Mercedes Bianco et al reports a case study of a pancreatic neuroendocrine tumour patient diagnosed with the tuberculosis after taking Everolimus medication. The authors further performed in vitro analysis to look at the root cause reasons. This manuscript is well structured overall, but it needs minor modifications, that I have outlined in the comments below.

We thank the reviewer for the overall positive.

In the point-by-point rebuttal, we address the specific points raised by the reviewer.

  1. Introduction section is very concise, please elaborate.

We thank the reviewer for the suggestion. We have revised the introduction section, elaborating the first paragraph.

  1. Line 41-42: The sentence is incomplete and does not make sense to the readers. Please correct them.

We thank for the suggestion, and we have corrected the text in the revised manuscript.

  1. Line 45-48: Please add reference to this sentence.

We thank for the suggestion, and we have included the references in the revised manuscript.

  1. Line 83: The authors have to explain in detail why the cancer drugs cisplatin, etoposide and FOLFIRI was stopped in 2016 and started a new drug Everolimus?

We thank the reviewer for the comment. Since the patient was referred to our centre only in august 2018, we lack details on therapeutic regimens. We have added a sentence in the M&M section to address this point.

  1. Line 90: If Everolimus drug was suspended, what other anticancer drug was given to the patient for pancreatic neuroendocrine tumour?

We thank the reviewer for the comment. The patient as followed by the infectious diseases team of our institution, while his oncology team was based elsewhere. He did not mention any other anti-cancer treatment at the follow-up visits in the period between August and November 2018. It is not unusual to suspend antineoplastic treatments while a concomitant acute health issue that might be negatively affected by them is present. Moreover, as the patient was restarted on everolimus due to disease progression in November it would be safe to assume he did not take any antineoplastic medication during this period. However, we preferred to add a sentence regarding in the case description section regarding lacking some details on therapeutic regimen. This would not be relevant to study as any medication taken in this period would not affect TB development as the patient would have had a positive sputum sample previous to starting the intervention.

  1. What dosage of Everolimus drug was given to the patient? Did the authors used the same concentration for the in vitro studies?

We thank the reviewer for his/her observation. Therapeutic drug monitoring of everolimus is not routinely performed in oncology (Deppenweiler et al., 2017) so despite knowing the dosage of everolimus given to the patient we can only estimate the concentration of the drug in the patient’s systems, from data in the literature this correspond to the maximal concentration used in this study,  as we have reported in the revised version of the manuscript. The other used concentration were chosen based on the literature 1ng corresponds to the concentration used in Ashley et al. (Ashley et al., 2020) and 3 and 8 ng are the to extremes of everolimus therapeutic range in kidney transplant.

  1. Please add references to the section 2.2.2.

We thank the reviewer for his/her observation. We have included the reference in the revised manuscript.

  1. Line 173-177: To look at the broader prospect, why the authors haven’t included another Asian Mtb strains as a control?

We apologize for not stating clearly this point and we thank the reviewer for the observation.  Actually, we have specifically selected, among the Mtb clinical isolated previously characterized, an Mtb strain belonging to the superlineage 1 EAI, that is in fact the ancient Asian lineage, to use it as a control for the Mtb GM2659 isolated from the patients and belonging to the same lineage. We have stated these important points in paragraph 2.3.

  1. Please add conclusion section.

We thank the reviewer for his/her comment, and we have modified accordingly to the suggestion.

  1. The authors have to proofread the manuscript, there are many grammatical errors.

We thank the reviewer for the suggestion. We have revised the manuscript and corrected the grammatical errors.

This manuscript is a resubmission of an earlier submission. The following is a list of the peer review reports and author responses from that submission.

Round 1

Reviewer 1 Report

This study mainly evaluated the direct and indirect activity of Everolimus on Mtb H37RV reference strain and the clinical GM2659 strain. Also,the authors quantitatively analyzed the autophagic flux of different Mtb strains with or without Everolimus by confocal pH-imaging of autophagic intermediates.They concluded that Everolimus did not exert a direct or indirect activity on Mtb as well as a negligible effect during Mtb infection in host cells despite stimulating autophagy and ROS production.The results provides a new perspective on the anti-mycobacteial effects of Everolimus.However,there are some problems to be further improved as well:

1, Please add the discussion on the possible reseason of different results reported in the current study and Ref 7.

2,The references could be improved.The author,for instance,incorrectly cites the reference [18] that is not related to Mtb Erdman on page 8,line 269-270.

3,If it is possible,Figure 3A should be made bigger for the reader’s convenience.Also,it would be better to add information of Figure 3C and 3D in figure legend.

4,In materials and methods part,4.2 and 4.6 has the same subtitle.

Author Response

Reviewer 1:

This study mainly evaluated the direct and indirect activity of Everolimus on Mtb H37RV reference strain and the clinical GM2659 strain. Also, the authors quantitatively analyzed the autophagic flux of different Mtb strains with or without Everolimus by confocal pH-imaging of autophagic intermediates. They concluded that Everolimus did not exert a direct or indirect activity on Mtb as well as a negligible effect during Mtb infection in host cells despite stimulating autophagy and ROS production. The results provide a new perspective on the anti-mycobacteial effects of Everolimus. However, there are some problems to be further improved as well:

We thank the reviewer for the overall positive comment and the interesting insights to improve the whole manuscript. In the point-by-point rebuttal, we address the specific points raised by the reviewer.

1, Please add the discussion on the possible reason of different results reported in the current study and Ref 7.

We thank the reviewer for his/her suggestion. We have included a sentence in the discussion section to explain our different findings about Everolimus treatment as reported: “The lack of activity of Everolimus, in the experimental models used in this study, against Mtb strains belonging to modern and ancient lineages suggests that more studies are needed before including Everolimus in the drugs that may be repurposed as HDTs against TB [1])”

2,The references could be improved. The author,for instance,incorrectly cites the reference [18] that is not related to Mtb Erdman on page 8,line 269-270.

We thank the reviewer for his/her comment, and we apologize for the mistake. In the revised manuscript we have checked and included the corrected reference.

3,If it is possible, Figure 3A should be made bigger for the reader’s convenience. Also,it would be better to add information of Figure 3C and 3D in figure legend.

We thank the reviewer for his/her observation. As suggested, we have also increased the size of the panel A and we have now included in the figure legend the information about panels C and D.

4,In materials and methods part,4.2 and 4.6 has the same subtitle.

We thank the reviewer for his/her comment. We have revised the manuscript and corrected the two subtitles.

Reviewer 2 Report

The manuscript entitled 'Evaluation of the Everolimus activity against Mycobacterium tuberculosis using in vitro models of infection' by Bianco et al. investigated the effect of Everolimus on drug sensitive Mtb clinical strain including its viability in an axenic culture and in a Peripheral Blood Mononuclear Cells (PBMCs) infection model. The authors also studied the effect of Everolimus treatment on ROS production and induction of autophagy during Mtb infection.

However, authors failed to provide detailed interpretation of the study. More importantly the study hypothesis and experimental results do not go hand in hand and the study also did not provide any evidences of functional relevance of Everolimus.

In this study, since, Evarolimus did not significantly induce autophagy and ROS

production, so, authors should not emphasize on this aspect (Line 28-29, 291-92). Authors are advised to perform more experiments in order to confirm the findings and also look for possible other mode of actions if not autophagy or ROS.

In lines 288-89, authors claimed that ‘mycobacteria can inhibit the late phases of autophagy progression, with early autophagosomes serving as a protected niche for replication’. Although, no conclusive experimental evidences are shown to support the claim in this study.

Data in figures is too less to draw any conclusion and experiments are also fewer without concrete data analysis.

Author Response

Reviewer 2:

The manuscript entitled 'Evaluation of the Everolimus activity against Mycobacterium tuberculosis using in vitro models of infection' by Bianco et al. investigated the effect of Everolimus on drug sensitive Mtb clinical strain including its viability in an axenic culture and in a Peripheral Blood Mononuclear Cells (PBMCs) infection model. The authors also studied the effect of Everolimus treatment on ROS production and induction of autophagy during Mtb infection.

However, authors failed to provide detailed interpretation of the study. More importantly the study hypothesis and experimental results do not go hand in hand and the study also did not provide any evidences of functional relevance of Everolimus.

We thank the reviewer for this observation, and we apologize as we did not explicitly state the aim of our study in the manuscript. As outlined in the manuscript, our study originates from a clinical case with the identification of an active TB disease in a patient receiving treatment with Everolimus. Following a review of the literature, we aimed at testing experimentally the direct and indirect activity of Everolimus against Mtb. We have included a sentence to better underline our aim as follow: “We do not aim to provide evidence of functional relevance of Everolimus, but rather to contribute to investigate its activity as HDTs against Mtb.

In this study, since, Everolimus did not significantly induce autophagy and ROS production, so, authors should not emphasize on this aspect (Line 28-29, 291-92). Authors are advised to perform more experiments in order to confirm the findings and also look for possible other mode of actions if not autophagy or ROS.

We thank the reviewer for the observation. We will modify lines 28-29. Regarding lines 291-92 we are describing the proposed mechanisms by which Everolimus may exert an antimycobacterial effect according to other authors. We agree that Everolimus can modulate several cellular processes and we are aware that when autophagy is implicated the effects are multiple and difficult to address. Given this complexity, we have focused our study on the most likely and conceivable mechanisms involving the role of Everolimus against Mtb infection, based also on previous observations. As suggested, we have revised the text and included a sentence remarking this concept, as follows: Everolimus has been proposed to exert a strong anti-mycobacterial activity in axenic culture and during Mtb infection of PBMCs by regulating autophagy and inducing ROS production[2] and “Finally, further studies may be performed to assess the impact of Everolimus, and more in general all HDTs, investigating the host immune system response due to the multiplicity and different cell processes that these treatments can directly or indirectly modulate, especially the mechanisms associated with the autophagy, which remain still complex, elusive and enigmatic”.

For the reasons provided in the previous and following points, it would be extremely challenging to investigate all the potential mechanism involved in the observed “lack of activity” of Everolimus.

In lines 288-89, authors claimed that ‘mycobacteria can inhibit the late phases of autophagy progression, with early autophagosomes serving as a protected niche for replication’. Although, no conclusive experimental evidences are shown to support the claim in this study.

We thank the reviewer for the comment. We refer to previous experimental evidence obtained by our group showing the innate capability of Mtb to evade macrophage intracellular killing by manipulating the autophagic process [3], [4]. It is not the scope of this manuscript to investigate the mechanisms of immune evasion deployed by Mtb to manipulate the processes associated with autophagy, which are very complex and still enigmatic (see for instance Kimmey et al Nature 2015[5]). The finding that activation of the autophagic process does not necessarily leads to improved Mtb killing [4], least in part explain the lack of activity against Mtb of Everolimus that we observed in this study.

Data in figures is too less to draw any conclusion and experiments are also fewer without concrete data analysis.

We would like to stress that this work stems from a clinical observation that led us to inquire experimentally on the potential usefulness of Everolimus against Mtb. In our opinion, experimental results like those provided in this manuscript (carried out in both CFUs or autophagy investigation) contribute to understand and better ponder the potential application of Everolimus against Mtb infection.
